# Sex Differences in Salmonellosis Incidence Rates—An Eight-Country National Data-Pooled Analysis

**DOI:** 10.3390/jcm10245767

**Published:** 2021-12-09

**Authors:** Victoria Peer, Naama Schwartz, Manfred S. Green

**Affiliations:** School of Public Health, University of Haifa, Haifa 3498838, Israel; naama.stat@gmail.com (N.S.); manfred.s.green@gmail.com (M.S.G.)

**Keywords:** salmonellosis, sex differences, hormones, incidence rates

## Abstract

Background: There are few studies on sex differences in the incidence rates (IR) for salmonellosis over several countries by age and time period. The purpose of this study was to explore the extent and consistency of the sex and age-specific differences. Methods: We analyzed national data from eight countries between 1994 and 2016. We computed country-specific male to female incidence rate ratios (IRRs) for each age group and pooled the data using meta-analytic methods. Variations of the IRRs by age, country and time period were evaluated using meta-regression. Results: The pooled male to female incidence RRs for ages 0–1, 1–4, 5–9 and 10–14, were 1.04 (1.02–1.06), 1.02 (1.01–1.03), 1.07 (1.05–1.08) and 1.28 (1.23–1.33), respectively. For the ages 15–44 and 45–64, the incidence rates were significantly higher in females. Meta-regression analyses indicate that age groups contributed most of the variation in the male to female IRRs. Conclusions: We suggest that genetic and hormonal factors and interactions between hormones and gut microbiota could contribute to the sex differences observed in young children. These findings should provide clues about the mechanisms of the infection, and should be useful in targeting treatments and development of vaccines. Highlights: (1) This manuscript provides consistent estimates of the excess salmonellosis incidence rates in male children up to age 15, which suggests an impact of sex hormones or genetic differences. (2) Our findings should promote the further investigations on sex-related determinants of infectious diseases.

## 1. Introduction

Salmonellosis is a common foodborne disease that affects the intestinal tract and most often presents as acute gastroenteritis. Humans become infected most frequently through contaminated water or food. Pathogens belonging to the Salmonella genus are facultative Gram-negative anaerobes, typically live in animal and human intestines and may shed through feces [1,2]. Within the Salmonella genus, *Salmonella enterica* is subdivided into six subspecies and includes more than 2500 serovares. The majority of the Salmonella strains that cause infection and represent a significant concern belong to the *Salmonella enterica* species. Salmonella has been identified as a commonly detected infection and the main cause of high morbidity and mortality in developing countries [1,2].

The observation that males appear to be more affected by COVID-19 than females once again emphasizes the importance of examining sex differences in infectious diseases [3]. Information on sex differences in salmonellosis incidence may provide guidance on the mechanisms of immune response to the infection. Studies on sex differences in salmonellosis incidence rates usually focus on individual countries, restricted to selected age groups and time periods [4,5,6,7,8,9,10]. The main goal of this study was to examine the extent and consistency of sex- and age-specific differences in salmonellosis over countries and years.

## 2. Materials and Methods

### 2.1. Search Strategy

We limited our search to published studies or through direct contact with national health organizations to obtain national data on salmonellosis incidence rates disaggregated by age and sex. We identified eight countries with well-established reporting and surveillance systems, with advanced health system and diagnostic facilities. The reporting system and the diagnostic methods of salmonellosis may be different over countries and time, however, there is no reason to believe that they differ by sex.

### 2.2. Sources of Data

#### 2.2.1. Source of Data

We managed to access detailed data on reported cases of salmonellosis, by age, sex and year, from a number of national institutions and surveillance systems, for eight countries. For Australia, data were obtained from the National Notifiable Diseases Surveillance System (NNDSS) for the years 2001–2016 [11], for Canada, from the Public Health Agency of Canada (PHAC) for 1991–2015 [12], for the Czech Republic, from the Institute of Health Information and Statistics for the years 2008–2013 [13], for Finland, from the National Institute for Health and Welfare (THL) for 1995–2016 [14], for Germany, from the German Federal Health Monitoring System for 2001–2016 [15], for New Zealand, from the Institute of Environmental Science and Research (ESR) for the Ministry of Health for 1997–2015 [16], for Poland, data for the years 2006–2016 were obtained from the National Institute of Public Health [17], and for Spain, from the Spanish Epidemiological Surveillance Network for 2005–2015 [18].

Data about the population denominators (disaggregated by age, sex and year) for the Australian population were obtained from the Australian Bureau of Statistics [19], for Canada from Statistics Canada [20], for the Czech Republic from the Czech Statistical Office [21], for Finland from the Statistics Finland’s PX-Web databases [22], for Germany from the German Federal Health Monitoring System [23], for New Zealand from Statistics New Zealand [24], for Poland from official website Statistics Poland [25] and for Spain from the Demographic Statistics Database [26]. 

#### 2.2.2. Ethics

National, anonymous data were published by official representatives of each country and there was no need for informed consent and research ethics board approval.

### 2.3. Statistical Analyses

#### 2.3.1. Calculation of Incidence Rates

Salmonellosis incidence rates (IR) for males and females, for each age group and country, for several years were calculated. Incidence rates per 100,000 of the population of the same sex were obtained as the number of published cases divided by the population size. The grouping by age was performed as follows: infants (<1), early childhood (1–4), late childhood (5–9), puberty (10–14), young adulthood (15–44 or 15–39), middle adulthood (45–64 or 40–59) and senior adulthood (65+ or 60+). The reporting systems of Canada, Finland and New Zealand used different age groups for the young adulthood (15–39), middle adulthood (40–59) and senior adulthood (60+). The male:female salmonellosis incidence rate ratio (RR) was obtained by dividing the male IR by the female IR.

#### 2.3.2. Meta-Analyses

Statistical analyses were performed using meta-analytic methods. 

The outcome variable in the study was the male:female salmonellosis IRR. Meta-analyses were performed on the subgroups of age, country and years. Pooled IRRs were calculated for each age subgroup, for all eight countries and time periods. The results are displayed in forest plots. Cochran’s Q test was used to determine the heterogeneity. Between-study variance was evaluated by Tau^2^ and I^2^. If I^2^ ≥ 50% and/or the Q test produced a *p*-value < 0.1, the random effects model was performed to measure pooled RRs and CI (95% Confidence Interval). To assess the impact of each county or years on the pooled male:female IRRs, we conducted sensitivity analysis and re-evaluated the pooled RRs for each age subgroup. We used meta-regression to identify the source of variation in the IRRs, including age group, country and group of years. The meta-analytic methods and meta-regressions were performed using STATA software version 12.1 (Stata Corp., College Station, TX, USA). The Egger test (illustrated as funnel plots) for asymmetry was performed for exploring possible imbalance in the impact of countries or group of years.

## 3. Results

### Descriptive Statistics

The summary of salmonellosis incidence rates (by sex, per 100,000 population) in different countries for each age subgroup is presented in Table 1. In the Czech Republic at ages <1–14, the incidence rates of salmonellosis were highest compared to other countries. In Spain, at ages 15–65, salmonellosis incidence rates for both sexes were lower than in all other countries.

The results of the meta-analyses by age groups are displayed as forest plots (Appendix A). The total male:female RR at age < 1 was 1.04 (95% CI 1.02–1.06, Appendix A) with low heterogeneity (I^2^ = 10.8% and Tau^2^ = 0.0004). The IRRs varied from 1.02 in the Czech Republic to 1.06 in New Zealand, Poland and Spain.

The forest plot for age 1–4 is presented in Appendix A. The total male:female RR = 1.02 (95% CI 1.01–1.03), with I^2^ = 29.7% and Tau^2^ = 0.0003, and varied from 1 in Germany to 1.05 in Canada.

The forest plot for ages 5–9 is presented in Appendix A. The pooled RR was 1.07 (95% CI 1.05–1.08), with I^2^ = 26.8% and Tau^2^ = 0.0006. Subgroup pooled IRRs by country varied from 1.01 in Poland to 1.14 in New Zealand.

The forest plot for age group 10–14 is presented in Appendix A. The total male:female incidence RR was 1.28 (95% CI 1.23–1.33), with I^2^ = 79.1% and Tau^2^ = 0.0108. The sub-group RRs by country varied from 1.12 in Poland to 1.61 in New Zealand.

The forest plot for age for young adulthood is presented in Appendix A. The pooled incidence RR = 0.91 for all countries together (95% CI 0.89–0.93), with I^2^ = 92.5% and Tau^2^ = 0.068. The subgroup RRs by country ranged from 0.78 for the Czech Republic to 1.08 for New Zealand.

The forest plot for middle adulthood is presented in Appendix A. The total male:female incidence RR = 0.89 (95% CI 0.86–0.92), I^2^ = 92.8% and Tau^2^ = 0.0121, with RRs ranging from 0.68 in the Czech Republic to 1.16 in Spain.

The forest plot for senior adulthood is presented in Appendix A. The pooled incidence RR was 1.03, 95% CI 1–1.07, I^2^ = 89.5% and Tau^2^ = 0.0109. The subtotal RRs by country varied from 0.83 in the Czech Republic to 1.35 in Spain.

In order to identify factors that may have an unduly influence on the pooled male:female incidence RRs, leave-one-out sensitivity analysis was used. 

One country or year period’s omission at a time caused only slight changes in the total male:female incidence RRs (Table 2 and Table 3, respectively). Data from the Czech Republic indicate a high IR of salmonellosis cases. Sensitivity analysis determined that the data from the Czech Republic did not affect overall results by age groups. 

Meta-regression analyses revealed that age groups contributed most of the differences in the male:female incidence RRs. For age < 1, the IRR was higher than in young and middle adulthood (*p* < 0.0001), but lower than in puberty (*p* < 0.0001). The incidence RRs for ages 1–4 and 5–9 were also significantly higher than for ages 15–39/44 and 40–59/45–64 (*p* < 0.0001). The incidence RRs for puberty were higher than in young, middle and senior adulthood (*p* < 0.0001). There was no association of incidence RRs for reported years.

The *p*-value for asymmetry (Egger’s test) was significant for early childhood and puberty, *p* = 0.095 and *p* = 0.001, respectively. Data regarding the asymmetry test for all age groups are presented in Figure 1**.**

## 4. Discussion

In this study, we studied the sex differences in salmonellosis incidence rates in eight countries by age group over a period of 6 to 22 years. Our results show a consistent excess in disease incidence rates in males under the age of 15. In older ages, in early and middle adulthood, there is an excess in females. In the oldest group, 65 and over, there is again an excess in males.

This study has a number of strengths and limitations. The strengths lie in the use of national incidence rates of reported cases from eight countries over a number of years. This should avoid some of the selection bias in smaller studies. In order to understand the magnitude of the findings in all countries that represent the population of Europe, Canada and Australia/New Zealand, we focused on countries with developed health systems. In these countries, there is no reason to assume that there is selective treatment or attitude according to the sex. For example, the study on the United States population relates to the issue and shows the identical medical care for salmonellosis for both sexes [8]. There may be heterogeneity in national reporting systems as well as the clinical criteria and laboratory-confirmed cases, but these factors should not differ between sexes. Information bias should be taken into consideration, especially because of non-specific clinical performance of the disease or the lack of laboratory confirmation or under-reporting in all reported countries. We assume that, even if this is the case, there is a low likelihood for it to be different between the sexes. Differences between males and females has been reported in the incidence of bacterial diseases, including in children [27]. Previous studies on the incidence of salmonellosis have focused on a particular country or relied on information from medical institutions [4,5,6,7,8,9,10], which could introduce bias. Most studies did not address the size of the population (as a denominator) of males or females and the findings were inconsistent.

Sex differences in the incidence of salmonellosis have been observed in a number of studies. In a study on 116,362 isolates of non-typhoidal Salmonella from England, between years 2004 and 2015, there were more males (unadjusted OR, 1.46; 95% CI, 1.35 to 1.59) [4]. Non-typhoidal Salmonella detected in blood in the United States, in 2003–2013, was associated with male sex and adults ≥ 65 years [5]: 272 adults with non-typhoidal Salmonella bacteremia were included in the study and males predominated (58.5%, with age range of 19–98 years, and a median age of 63 years) [6]. In Nairobi, Kenya, 332 children aged between 1 month and 7 years with salmonellosis recruited in the study showed no difference between sexes (*p* > 0.05) [7]. In a study that was conducted in the United States over a 33-year period (1968–2000), there was an excess of salmonellosis in adult women [8]. Surveillance data on 11,243 subjects older than 18 years collected over 3 years from two hospitals in Shanghai [9] showed that Salmonella infection was similar between males and females (50.1% and 47.5%). In a study conducted by GBD 2017, non-Typhoidal Salmonella Invasive Disease Collaborators, on 535,000 cases that occurred in the years 1990 to 2017, incidence of Salmonella was not significantly different between sexes in all ages [10].

We hypothesized that the sex differences in salmonellosis incidences include exposure factors, biological (genetic and hormonal differences) factors and interactions between hormones and gut microbiota. As regards exposure, outbreak investigations have shown that the majority of salmonellosis infections result from consuming Salmonella-contaminated foods [28]. We hypothesized that the main cause of female predominance (in young and middle adulthood groups) in Salmonella incidence is excessive exposure. In general, females consume more fruits and vegetables than males [29]. The differences between sexes in eating habits [30,31] and overall exposure influence salmonellosis incidence. Females’ greater exposure to poultry during food-handling practices [32] and childcare place adult women (especially in young and middle adulthood) at increased risk of salmonellosis.

We suggest that the heterogeneity over time is likely a consequence of random variation. The heterogeneity in puberty and in older ages may result from random variation in the Salmonella male:female RR between reported year groups and between countries.

Biological differences between sexes should not be ignored. Biological and hormonal sex differences can lead to more robust immune responses in females and to differences in disease outcomes between sexes due to infections [33]. However, sex-specific differences can provide an additional biological mechanism mediating differences in microbiota composition. Healthy females have greater pro-inflammatory genes’ expression in gastrointestinal tract samples compared with males [34]. It is possible that gut inflammatory status influences the sex-specific pattern in healthy and asymptomatic individuals and may contribute to female immune systems’ overreaction. These differences between male and female microbial composition, in part driven by sex hormones, may affect intestinal inflammatory disease outcomes and severity [35,36].

There do not appear to be any reasons for suspecting sex differences in exposure in infants and young children. Contact with live animals can result in salmonellosis in childhood and among children aged < 1 year, but the exposure is not likely to differ between the sexes and it seems to be a marginal factor to explain the male predominance in disease incidence [37].

We hypothesized that the higher salmonellosis incidence rate in young males under the age of 15 may be explained by genetic and hormonal factors. Both sex hormones and the X chromosome [38] may contribute to the immune response even in young age groups. The differential responses of immune cells to pathogens in the presence of hormones contribute to explaining age and sex differences in immune responses and in diseases [38]. In infants, the short activation of the pituitary-gonadal hormone axis is defined as “mini-puberty” [39]. Higher estrogen levels are found in girls even before physical manifestation of pubertal maturation [40].

Sex hormones, mainly estrogen, regulate immune responses, cytokine production and induction [33]. High levels of IFN-γ, TNF-α and the inflammatory interleukins IL-18, IL-12 and IL-15 participate in Salmonella clearance [41,42], along with CD4+ T cells, the key factor mediating immunity against the Salmonella pathogen via the IFN-γ production. A variety of estrogen-regulated cytokines that are produced as a result of immune response provide the appropriate environmental conditions for CD4+ T cells’ differentiation towards both Th1 and Th17 cells that are essential for the immune response to Salmonella [43].

As hypothesized, sex hormones lead to gut estrogen-regulated microbiota composition differences between girls and boys, even in early life [44]. Since the composition of the human gut microbiota is regulated by estrogen, we can assume that differences in the gut microbiota may be a factor of the sexual dimorphism in salmonellosis [45,46].

## 5. Conclusions

Future research should be performed in order to understand the patterns and pathways that mediate the immune system response and impact immunity to infectious diseases differently in males compared with females. The consistent sex differences observed between males and females in the incidence of Salmonellosis emphasizes the need to consider “sex” as a biological variable in studies of communicable infectious diseases and public health.

## Figures and Tables

**Figure 1 jcm-10-05767-f001:**
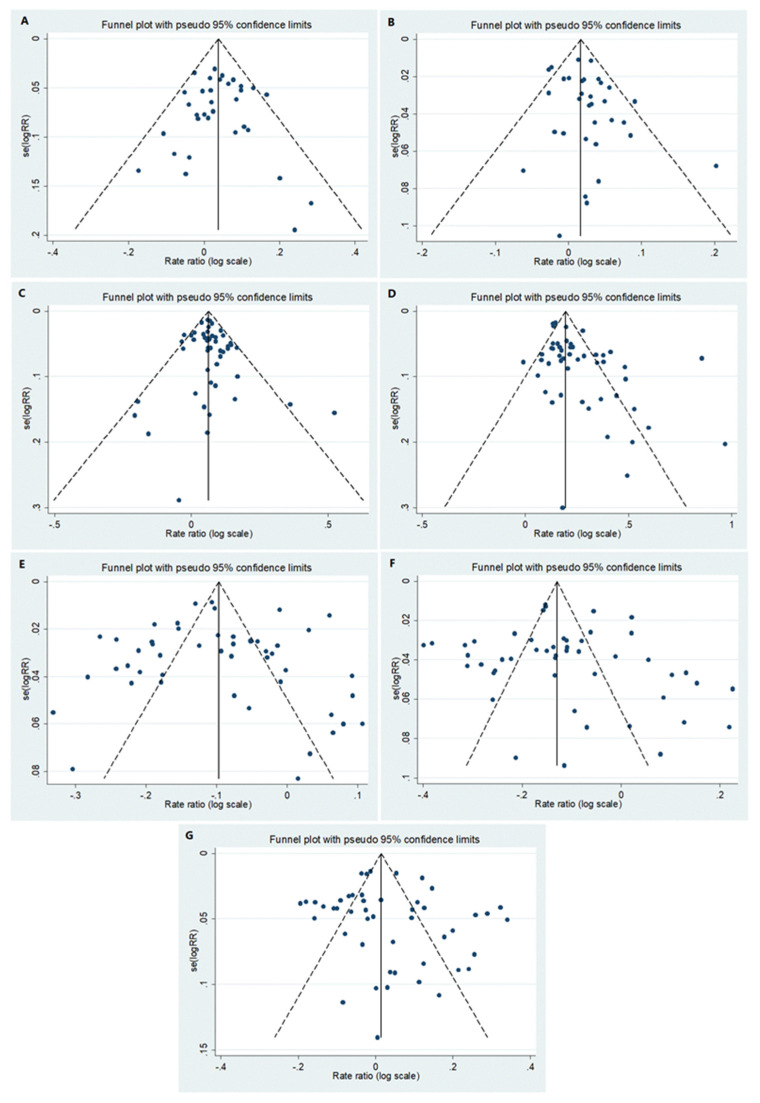
Funnel plots for the following age groups: (**A**) <1, (**B**) 1–4, (**C**) 5–9, (**D**) 10–14, (**E**) 15–39/15–44, (**F**) 40–59/45–64 and (**G**) 60+/65+.

**Table 1 jcm-10-05767-t001:** Descriptive data: details of the countries included in the analysis, disaggregated by sex and age.

			Males		Females	
Age Group	Country	Years	*n/N*	IR Males	*n/N*	IR Females
<1	Canada	1994–2015	3196/4,066,314	78.6	2895/3,861,381	75
	Czech Republic	2008–2013	1463/349,195	419	1358/332,712	408.2
	Germany	2001–2016	7653/5,740,478	133.3	7179/5,448,550	131.8
	New Zealand	1997–2015	840/576,900	145.6	758/548,520	138.2
	Poland	2006–2016	4602/2,177,523	211.3	4092/2,055,764	199.1
	Spain	2005–2015	2769/2,679,186	103.4	2423/2,514,548	96.4
1–4	Canada	1994–2015	9099/16,718,349	54.4	8210/15,900,004	51.6
	Czech Republic	2008–2013	8536/1,410,748	605.1	7966/1,343,670	592.9
	Germany	2001–2016	55,616/23,509,315	236.6	52,459/22,311,030	235.1
	New Zealand	1997–2015	2727/2,308,880	118.1	2486/2,191,980	113.4
	Poland	2006–2016	18,051/8,763,810	206	16,528/8,294,052	199.3
	Spain	2005–2015	9130/10,880,587	83.9	8516/10,233,932	83.2
5–9	Australia	2001–2016	6517/11,398,585	57.2	5877/10,814,642	54.3
	Canada	1994–2015	5868/21,678,340	27.1	5045/20,622,712	24.5
	Czech Republic	2008–2013	4685/1,532,669	305.7	4067/1,450,621	280.4
	Finland	1995–2016	1035/3,440,956	30.1	956/3,297,629	29.0
	Germany	2001–2016	39,375/30,760,941	128	35,115/29,187,252	120.3
	New Zealand	1997–2015	1115/2,899,540	38.5	925/2,752,910	33.6
	Poland	2006–2016	7721/10,753,278	71.8	7268/10,206,501	71.2
	Spain	2005–2015	4725/13,017,097	36.3	4019/12,287,011	32.7
10–14	Australia	2001–2016	4433/11,377,822	39	3315/10,797,396	30.7
	Canada	1994–2015	4066/22,713,799	17.9	2891/21,572,803	13.4
	Czech Republic	2008–2013	1813/1,416,001	128	1483/1,339,518	110.7
	Finland	1995–2016	1115/3,522,497	31.7	841/3,375,446	24.9
	Germany	2001–2016	24,832/33,455,166	74.2	19,955/31,724,889	62.9
	New Zealand	1997–2015	790/2,919,850	27.1	466/2,776,650	16.8
	Poland	2006–2016	2805/11,130,177	25.2	2384/10,591,951	22.5
	Spain	2005–2015	1750/12,301,238	14.2	1109/11,627,137	9.5
15–44	Australia	2001–2016	27,906/73,591,102	37.9	31,662/72,741,755	43.5
	Canada	1994–2015	22,168/126,619,246	17.5	22,933/123,505,034	18.6
	Czech Republic	2008–2013	6462/13,725,818	47.1	7878/12,978,912	60.7
	Finland	1995–2016	10,300/18,898,064	54.5	12,418/18,050,351	68.8
	Germany	2001–2016	94,718/257,895,408	36.7	97,495/247,590,330	39.4
	New Zealand	1997–2015	4270/13,546,700	31.5	4093/13,976,900	29.3
	Poland	2006–2016	9128/92,802,239	9.8	10380/90,097,352	11.5
	Spain	2005–2015	4401/110,542,308	4	4200/105,413,400	4
45–64	Australia	2001–2016	12,790/41,988,401	30.5	14,697/42,573,071	34.5
	Canada	1994–2015	13,462/100,585,696	13.4	15,213/99,821,361	15.2
	Czech Republic	2008–2013	3233/8,403,729	38.5	4905/8,624,880	56.9
	Finland	1995–2016	8577/16,513,241	51.9	11,169/16,307,550	68.5
	Germany	2001–2016	51,368/181,698,132	28.3	57,083/181,849,520	31.4
	New Zealand	1997–2015	2572/10,201,030	25.2	2525/10,685,350	23.6
	Poland	2006–2016	5532/55,127,862	10	7126/59,180,678	12
	Spain	2005–2015	3449/63,103,755	5.5	3031/64,340,310	4.7
65+	Australia	2001–2016	8088/21,417,772	37.8	10,239/25,538,457	40.1
	Canada	1994–2015	9190/58,764,646	15.6	11,888/70,995,360	16.7
	Czech Republic	2008–2013	2212/4,087,584	54.1	3914/5,999,018	65.2
	Finland	1995–2016	2572/11,159,619	23	3161/15,066,114	21
	Germany	2001–2016	39,020/108,019,284	36.1	53,965/149,862,231	36
	New Zealand	1997–2015	1410/6,302,700	22.4	1524/7,386,000	20.6
	Poland	2006–2016	4110/23,115,840	17.8	6148/37,363,573	16.5
	Spain	2005–2015	3834/37,127,234	10.3	3795/49,879,431	7.6

IR = incidence rate, IR per 100,000 Male or Female populations. *n*—Cumulative number of Salmonella cases for given years. *N*—Cumulative number of the population for given years.

**Table 2 jcm-10-05767-t002:** Sensitivity analysis, by age group and country.

Age Group							
Country Removed	Infants RR (CI)	Early Childhood RR (CI)	Late Childhood RR (CI)	Puberty RR (CI)	Young Adulthood RR (CI)	Middle Adulthood RR (CI)	Senior Adulthood RR (CI)
Australia	-	-	1.08 (1.04–1.11)	1.29 (1.19–1.39)	0.9 (0.85–0.96)	0.88 (0.81–0.97)	1.04 (0.96–1.14)
Canada	1.04 (1.01–1.07)	1.02 (1–1.03)	1.07 (1.04–1.09)	1.28 (1.19–1.37)	0.89 (0.84–0.95)	0.88 (0.81–0.97)	1.05 (0.96–1.14)
Czech Republic	1.04 (1.02–1.07)	1.03 (1.01- 1.05)	1.07 (1.04–1.1)	1.3 (1.21–1.4)	0.92 (0.87–0.97)	0.91 (0.85–0.98)	1.06 (0.98–1.15)
Finland	-	-	1.07 (1.05–1.1)	1.29 (1.2–1.38)	0.92 (0.87–0.96)	0.9 (0.84–0.97)	1.02 (0.94–1.11)
Germany	1.06 (1.03–1.08)	1.03 (1.02–1.05)	1.07 (1.04–1.11)	1.3 (1.2–1.41)	0.9 (0.84–0.96)	0.88 (0.79–0.97)	1.03 (0.93–1.15)
New Zealand	1.04 (1.02–1.07)	1.02 (1.01–1.04)	1.07 (1.04–1.09)	1.25 (1.18–1.33)	0.88 (0.83–0.93)	0.86 (0.8–0.93)	1.02 (0.94–1.11)
Poland	1.03 (1.01–1.06)	1.02 (1.003–1.04)	1.08 (1.06–1.1)	1.31 (1.22–1.41)	0.91 (0.85–0.96)	0.89 (0.82–0.97)	1.02 (0.94–1.11)
Spain	1.03 (1.01–1.06)	1.03 (1.01–1.05)	1.07 (1.04–1.09)	1.26 (1.18–1.33)	0.89 (0.84–0.94)	0.85 (0.79–0.91)	0.99 (0.94–1.04)

CI = confidence interval. RR = rate ratio.

**Table 3 jcm-10-05767-t003:** Sensitivity analysis, by age group and years.

Age Group							
Years Removed	Infants RR (CI)	Early Childhood RR (CI)	Late Childhood RR (CI)	Puberty RR (CI)	Young Adulthood RR (CI)	Middle Adulthood RR (CI)	Senior Adulthood RR (CI)
1994/1995–1996	1.04 (1.01–1.07)	1.02 (1.01–1.03)	1.07 (1.05–1.08)	1.22 (1.17–1.27)	0.91 (0.9–0.91)	0.88 (0.86–0.9)	1.01 (0.96–1.05)
1997–2000	1.03 (1.01–1.05)	1.02 (1.01–1.03)	1.07 (1.05–1.08)	1.23 (1.18–1.28)	0.91 (0.9–0.92)	0.88 (0.86–0.9)	1.01 (0.97–1.05)
2001–2002	1.04 (1.02–1.07)	1.02 (1.003–1.03)	1.06 (1.05–1.08)	1.24 (1.19–1.29)	0.91 (0.9–0.92)	0.88 (0.86–0.9)	1.01 (0.97–1.05)
2003–2005	1.04 (1.01–1.07)	1.02 (1.01–1.03)	1.07 (1.05–1.08)	1.23 (1.18–1.29)	0.91 (0.9–0.92)	0.88 (0.86–0.9)	1.01 (0.97–1.05)
2005/2006–2007	1.03 (1.004–1.06)	1.02 (1.01–1.03)	1.07 (1.06–1.08)	1.23 (1.18–1.28)	0.91 (0.9–0.92)	0.88 (0.85–0.9)	1.01 (0.96–1.05)
2008–2010	1.03 (1.003–1.06)	1.02 (1.01–1.03)	1.06 (1.05–1.08)	1.23 (1.17–1.28)	0.9 (0.9–0.91)	0.88 (0.85–0.9)	0.997 (0.95–1.04)
2011–2013	1.04 (1.01–1.07)	1.02 (1.01–1.03)	1.06 (1.05–1.08)	1.22 (1.17–1.27)	0.9 (0.9–0.91)	0.87 (0.85–0.89)	0.99 (0.95–1.04)
2014–2015	1.04 (1.01–1.07)	1.02 (1.006–1.03)	1.07 (1.05–1.08)	1.21 (1.17–1.24)	0.91 (0.9–0.92)	0.87 (0.85–0.89)	0.99 (0.95–1.03)
2016	1.04 (1.01–1.07)	1.02 (1.01–1.03)	1.07 (1.06–1.08)	1.23 (1.19–1.28)	0.91 (0.9–0.92)	0.87 (0.85–0.9)	0.99 (0.95–1.04)

CI = confidence interval. RR = rate ratio.

## Data Availability

All data are available from the original sources or from the authors. Links to publicly archived datasets analyzed or generated during the study are included in References.

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
