# Peer review of "Sex Differences in Salmonellosis Incidence Rates—An Eight-Country National Data-Pooled Analysis"

_jcm, 2021, doi:10.3390/jcm10245767_

Round 1

Reviewer 1 Report

The manuscript is very well written and the study is well designed. The study is well contextualized and the results very well discussed with appropriate references.

I kindly would like suggest minor modifications:

Introduction. Line 34 change to: Gram-negative (no italics) aerobic to facultatively anaerobic bacillus...

The abstract can be improved, it should not provide so many values and statistical details which do not really explain and valorize the work. Instead, you could describe the main findings.

Figures 1 to 7 are described in the text and could be presented as supplemental material.

Author Response

We thank you for the  review and suggestions.

Title: Sex differences in salmonellosis incidence rates - an eight- country national data pooled analysis

Authors: Victoria Peer *, Naama Schwartz, Manfred Green

Reviewer 1

  1. English language and style are fine/minor spell check required

Response: We've made changes in spelling in entire manuscript.

  1. Comments and Suggestions for Authors

The manuscript is very well written and the study is well designed. The study is well contextualized and the results very well discussed with appropriate references.

I kindly would like suggest minor modifications:

  • Line 34 change to: Gram-negative (no italics) aerobic to

           facultatively anaerobic bacillus.

Response: Done. This sentence is rewritten (lines 30-32:''Pathogens belonging to the Salmonella genus are facultative Gram-negative anaerobes, typically live in animal and human intestines and may shed through feces'').

  • The abstract can be improved, it should not provide so many values and statistical details which do not really explain and valorize the work. Instead, you could describe the main findings.

Response: We accept your comment. The abstract was rewritten            according to your suggestions.

  • Figures 1 to 7 are described in the text and could be presented as supplemental material.

Response: We accept your comment.  In the section of results, it was mentioned that figures (Fig1 to 7) would be a part of supplemental materials.

Reviewer 2 Report

The manuscript provides important finding on the sex diffrences in salmonellelosis.

Introduction

Line 34 change to 'facultative Gram-negative anaerobe'

Include one or two sentences on Salmonella as an important cause of invasive disease in some parts of the world with high case fatality.

Discussion

line 245 check this and correct 'in compare to male'

Author Response

We thank you for your useful comment and suggestion

Title: Sex differences in salmonellosis incidence rates - an eight- country national data pooled analysis

Authors: Victoria Peer *, Naama Schwartz, Manfred Green

Reviewer 2

  1. English language and style are fine/minor spell check required

Response: We've made changes in spelling in entire manuscript.

  1. 2. Comments and Suggestions for Authors

The manuscript provides important finding on the sex differences in salmonellosis.

  • Line 34 change to 'facultative Gram-negative anaerobe. Include one or two sentences on Salmonella as an important cause of invasive disease in some parts of the world with high case fatality.

Response: Done. This sentence is rewritten (lines 30-32:''Pathogens belonging to the Salmonella genus are facultative Gram-negative anaerobes, typically live in animal and human intestines and may shed through feces'').

We included the sentences on Salmonella as an important cause of high morbidity and mortality in developing countries (lines 33-36).

  • Line 245 check this and correct 'in compare to male'.

Response: This sentence is rewritten (lines 227-228).